# Comparison of IncK-*bla*_CMY-2_ Plasmids in Extended-Spectrum Cephalosporin-Resistant *Escherichia coli* Isolated from Poultry and Humans in Denmark, Finland, and Germany

**DOI:** 10.3390/antibiotics13040349

**Published:** 2024-04-10

**Authors:** Meiyao Che, Ana Herrero Fresno, Cristina Calvo-Fernandez, Henrik Hasman, Paula E. Kurittu, Annamari Heikinheimo, Lisbeth Truelstrup Hansen

**Affiliations:** 1National Food Institute, Technical University of Denmark, 2800 Lyngby, Denmark; meic@food.dtu.dk (M.C.); ccfe@food.dtu.dk (C.C.-F.); 2Department of Biochemistry and Molecular Biology, Faculty of Sciences, Campus Terra, Universidade da Santiago de Compostela (USC), 27002 Lugo, Spain; ana.fresno@usc.es; 3Reference Laboratory for Antibiotic Resistance, Statens Serum Institut, Artillerivej 5, 2300 Copenhagen, Denmark; henh@ssi.dk; 4Department of Food Health and Environmental Hygiene, Faculty of Veterinary Medicine, University of Helsinki, 00014 Helsinki, Finland; paula.kurittu@gmail.com (P.E.K.); annamari.heikinheimo@helsinki.fi (A.H.); 5Microbiology Unit, Finnish Food Authority, Mustialankatu 3, 00790 Helsinki, Finland

**Keywords:** plasmids, IncK, *bla*
_CMY-2_, extended-spectrum resistant (ESC), *Escherichia coli*

## Abstract

*Escherichia coli* carrying IncK-*bla*_CMY-2_ plasmids mediating resistance to extended-spectrum cephalosporins (ESC) has been frequently described in food-producing animals and in humans. This study aimed to characterize IncK-*bla*_CMY-2_-positive ESC-resistant *E. coli* isolates from poultry production systems in Denmark, Finland, and Germany, as well as from Danish human blood infections, and further compare their plasmids. Whole-genome sequencing (Illumina) of all isolates (*n* = 46) confirmed the presence of the *bla*_CMY-2_ gene. Minimum inhibitory concentration (MIC) testing revealed a resistant phenotype to cefotaxime as well as resistance to ≥3 antibiotic classes. Conjugative transfer of the *bla*_CMY-2_ gene confirmed the resistance being on mobile plasmids. Pangenome analysis showed only one-third of the genes being in the core with the remainder being in the large accessory gene pool. Single nucleotide polymorphism (SNP) analysis on sequence type (ST) 429 and 1286 isolates showed between 0–60 and 13–90 SNP differences, respectively, indicating vertical transmission of closely related clones in the poultry production, including among Danish, Finnish, and German ST429 isolates. A comparison of 22 ST429 isolates from this study with 80 ST429 isolates in Enterobase revealed the widespread geographical occurrence of related isolates associated with poultry production. Long-read sequencing of a representative subset of isolates (*n* = 28) allowed further characterization and comparison of the IncK-*bla*_CMY-2_ plasmids with publicly available plasmid sequences. This analysis revealed the presence of highly similar plasmids in ESC-resistant *E. coli* from Denmark, Finland, and Germany pointing to the existence of common sources. Moreover, the analysis presented evidence of global plasmid transmission and evolution. Lastly, our results indicate that IncK-*bla*_CMY-2_ plasmids and their carriers had been circulating in the Danish production chain with an associated risk of spreading to humans, as exemplified by the similarity of the clinical ST429 isolate to poultry isolates. Its persistence may be driven by co-selection since most IncK-*bla*_CMY-2_ plasmids harbor resistance factors to drugs used in veterinary medicine.

## 1. Introduction

Extended-spectrum cephalosporins (ESC) are classified as critically essential antimicrobials by the World Health Organization. Therefore, their use is restricted to the treatment of infections in humans and animals caused by multidrug-resistant (MDR) Gram-negative bacteria, particularly Enterobacteriaceae such as MDR *Escherichia coli* (*E. coli*) [1]. In Europe, the Antimicrobial Advice Ad Hoc Expert Group (AMEG) of the European Medicines Agency (EMA) has classified third- and fourth-generation cephalosporins as belonging to “Category B-Restrict” antibiotics, meaning their use in animals should be restricted and preferably be based on antimicrobial susceptibility testing to mitigate the risk to public health [2]. Resistance to ESCs has dramatically increased worldwide in this family of bacteria and is encoded predominantly by the extended-spectrum β-lactamase (ESBLs) and AmpC β-lactamase (AmpCs) genes [3,4]. The plasmid-mediated AmpC-like gene, *bla*_CMY-2_, which reduces susceptibility to aztreonam, cephamycin, and third-generation cephalosporins, is one of the most prevalent among *E. coli* strains [5]. In general, the gene *bla*_CMY-2_ is located on plasmids belonging to several incompatibility groups, including IncI1, IncA/C, IncF, and IncK [6,7,8,9,10]. 

The incidence of *bla*_CMY-2_-related ESC resistance in *E. coli* belonging to various multilocus sequence types (STs) is common in livestock across several European countries, while being rarely reported in *E. coli* from humans in Europe [11,12,13]. Previous studies observed that IncI1 and IncK plasmids carrying *bla*_CMY-2_ occurred in ST38, ST131, and ST117 ESC-resistant *E. coli* isolates from human, livestock, and meat products in Germany, as well as in the same *E. coli* STs from healthy urban dogs in France [14,15] Also, IncK and IncI plasmids harboring *bla*_CMY-2_ were identified in Finnish poultry farms [16], while IncA/C plasmids containing *bla*_CMY-2_ were detected in particular *E. coli* lineages in Swedish broiler flocks [17]. Surveillance of antimicrobial resistance in Denmark in 2015–2016 documented that *E. coli* isolates carrying *bla*_CMY-2_ in Danish broiler meat came from widely distributed STs, i.e., ST38, ST154, ST2309, while imported broiler meat isolates belonged to ST23, ST117, ST131, ST2040 [18]. A recent study from Denmark revealed a close phylogenetic relatedness among *E. coli* ST131 IncI1- *bla*_CMY-2_ plasmids carrying isolates from broilers and a patient with a bloodstream infection which highlights the risk of the potential zoonotic spread of these antimicrobial-resistant bacteria [19]. A novel ST429 ESC-resistant *E. coli* harboring *bla*_CMY-2_ was detected in the Danish surveillance of both domestic broilers and their meat, in imported broiler meat [18] as well as in our previous study, which revealed that *bla*_CMY-2_ on IncI (ST2040) or IncK (ST429) plasmids dominated in a survey of ESC-resistant *E. coli* from Danish poultry farms and slaughterhouses over the period 2015–2018 [20]. However, knowledge is currently lacking regarding the IncK-*bla*_CMY-2_ plasmids’ relatedness among different ESC-resistant *E. coli* STs, including ST429, in the poultry production chain in Denmark and other countries, and from human clinical cases.

Therefore, the objective of this study was to investigate the genetic diversity of IncK-*bla*_CMY-2_-positive ESC-resistant *E. coli* isolates (*n* = 46) and further to identify the similarity of IncK-*bla*_CMY-2_ plasmids (*n* = 28) from the poultry production chains in Denmark, Finland, and Germany as well as from Danish patients. Also, we determined the spread of ST429 ESC-resistant *E. coli* carrying *bla*_CMY-2_ globally and compared IncK-*bla*_CMY-2_ plasmids using publicly available databases. 

## 2. Results

### 2.1. Phenotypic Antimicrobial-Resistance Profiles

Results of MIC distributions are summarized in Table 1 for all ESC-resistant *E. coli* IncK-*bla*_CMY-2_ isolates. Specific results for each of the isolates are provided in Appendix A. A total of 15 resistance profiles were observed. Based on the epidemiological cut-off values (ECOFFs), all 46 isolates were classified as non-wild type (i.e., resistant to antibiotics) to ampicillin, cefotaxime, and ceftazidime. Moreover, 93.5%, 84.8%, and 71.7% of the ESC-resistant *E. coli* isolates were classified as resistant to sulfamethoxazole, tetracycline, and gentamicin. In addition, the proportion of isolates classified as non-wild type to trimethoprim, nalidixic acid, and ciprofloxacin ranged from 19.6% to 8.6%. For other antimicrobials, MICs above ECOFFs were less common to azithromycin (2.2%) and chloramphenicol (2.2%), and all isolates were wild type with respect to amikacin, colistin, and tigecycline. Overall, all ESC-resistant *E. coli* isolates showed multidrug resistance, as they exhibited resistance towards three or more antibiotics belonging to different classes [21].

### 2.2. High Prevalence of Antimicrobial-Resistance Determinants

The heatmap of the ResFinder results obtained from analysis of the WGS from the 46 ESC-resistant *E. coli* isolates is presented in Figure 1, detailing the acquired resistance genes and their predicted resistance phenotypes. A total of 18 different resistance genes were identified with isolates carrying multiple resistance genes (2–12 genes, five being the most common). The isolates were selected based on the presence of *bla*_CMY-2_, and this gene was identified in all of them. The gene *bla*_TEM-1B_ was also identified in nine (19.6%) isolates. The most abundant co-resistance observed among the isolates was aminoglycoside resistance (*n* = 44, 95.7%) followed by sulfonamide (*n* = 43, 93.5%) and tetracycline resistance (*n* = 39, 84.8%). Aminoglycoside resistance was associated with the presence of *aadA1* (*n* = 40, 87.0%), *aac (3)-Vla* (*n* = 31, 67.4%), *aph (3″)-Ib* and *aph (6)-Id* (*n* = 4, 8.7%), *aac (3)-IId* (*n* = 2, 4.3%), *aadA5* (*n* = 1, 2.2%), *aph (3′)-Ia* (*n* = 3, 6.6%). Sulfonamide resistance was encoded by either *sul1* (*n* = 34, 74.0%) or *sul2* (*n* = 9, 19.6%), while tetracycline resistance was encoded by *tet(A)* (*n* = 36, 78.3%) or *tet(B)* (*n* = 3, 6.6%).

### 2.3. Results from Conjugation Assays

The conjugation experiments showed that all 46 ESC-resistant *E. coli* isolates could transfer cefotaxime resistance to the sensitive *E. coli* MG1655 with PCR results confirming the presence of the *bla*_CMY-2_ gene in the transconjugants. These results indicate that the *bla*_CMY-2_ gene in all ESC-resistant *E. coli* isolates regardless of their ST-types was located on a conjugative plasmid (Table 2).

### 2.4. Pangenome Analysis

Pangenome analysis accounts for the genomic flexibility among the 46 IncK-*bla*_CMY-2_-positive ESC-resistant *E. coli* isolates by considering the core and pangenome structure of the isolates, as shown in Figure 2. The pangenome in Figure 2a consisted of 11,804 genes in total, which could be divided into core genes, soft-core genes, shell genes, and cloud genes. The core genome (3094 genes, 27.91%) made up less than one-third of the collective pangenome genes for the ESC-resistant *E. coli* isolates; in contrast, results from the Roary pipeline generated 7990 accessory genes in total showing large variability among the isolates. Of these genes, soft-core genes found in 95–99% of the isolates comprised 351 genes (3.17%). Shell genes (2339 genes, 21.10%) were detected in between 15 and 95% of the isolates. Cloud gene families (5300 genes, 47.82%) were represented in less than 15% of the isolates, highlighting the high genomic variability. The pangenome clustering tree in Figure 2b shows the relatedness among ST429 isolates from different sources and countries of origin vs. the other STs, e.g., 57, 162, and 1286 (see Table 2 for information on ST), found in a separate cluster. Figure 2c visualizes the presence and absence of annotated genes in each of the isolates, clearly showing the shared core genes and the range of shell and clouds genes. Further details are found in Appendix A.

### 2.5. Phylogenetic Tree Analysis

The SNP-based trees revealed close relationships among isolates belonging to ST429 and ST1286, respectively. Isolates were obtained from different sources, i.e., poultry, poultry meat, and humans, over a long period from 2012 to 2018. As shown in Figure 3a, ST429 isolates obtained in a Danish slaughterhouse (SA17021, SA17022) and one human clinical case (HBI01) were highly similar with only 4–7 SNPs, indicating possible transfer from poultry to humans. Notably, two ST429 isolates (SA17023, SA17024), collected from the same slaughterhouse in Denmark, belonged to the same subcluster as the three German isolates from 2012 and all seven Finnish isolates also obtained in 2017, which suggests the introduction of the ST subcluster from a common source and clonal transmission in the poultry production chain. Furthermore, German isolates from 2012 clustering with Danish poultry and one human clinical isolate from 2015–2017, with 51 SNPs between the German RL97 and the Danish SA17023, again point out the longevity of this ST429 in chicken production and possible common sources transcending country boundaries. All six ST1286 isolates in Figure 3b came from one slaughterhouse in Denmark between 2017 and 2018 and differed by SNPs ranging from 13 (SB17103 vs. SB18102) to 90 (SB18001 vs. SB18004), suggesting a temporal colonization or repeated introduction of ST1286 in the Danish slaughterhouse. The resulting SNP matrices of ST429 and ST1286 are shown in Appendix A.

The minimum spanning tree based on the 102 ST429 ESC-resistant *E. coli* isolates from this study (Danish isolates 2015–2017, *n* = 22) and Enterobase (*n* = 80, 2010–2021) is shown in Figure 4. Overall, our results demonstrated the worldwide spread of ST429 ESC-resistant *E. coli* over the past decade, suggesting a role for global trade in the dissemination of these isolates. Of the 102 isolates depicted in Figure 4, 84 were related to poultry production (Appendix A). The Danish ST429 *bla*_CMY-2_-positive isolates were mainly found in two closely related clusters with isolates collected from other countries, with allelic differences ≤ 15. To note, the Danish isolates were observed to be highly similar to isolates (at the center of the tree) originating from UK poultry between 2013 and 2016, suggesting the possibility that ST429 *bla*_CMY-2_-positive Danish isolates were initially imported from Britain. Interestingly, SA17023 (Appendix A) grouped into the same subcluster with the two Sweden poultry isolates ESC_RA0430AA (accession no. SRR11473342) and ESC_RA0434AA (accession no. SRR11473346) in 2016 with only three SNP differences, and PS16004 showed five allelic differences with one isolate ESC_UA5215AA (accession no. ERR5443272) from Spain poultry in 2016. Meanwhile, these two subclusters with SA17023 and PS16004 also included UK poultry isolates from 2015, indicating that the poultry parent birds of Denmark, Sweden, and Spain possibly were imported from Britain.

### 2.6. Plasmid Characterization

The short-read sequencing and subsequent hybrid assembly (long- and short-reads combined) of plasmids from a subset of 28 representative *E. coli bla*_CMY-2_ isolates belonging to nine STs revealed that all isolates harbored IncK plasmids carrying *bla*_CMY-2_, with predicted sizes ranging from 85,938 to 133,315 bp (Table 2). Also, all hybrid assembled IncK-*bla*_CMY-2_ plasmids were found to be in one circular contig. Resfinder confirmed that six IncK plasmids harbored only the *bla*_CMY-2_ resistance gene, i.e., plasmids pSB17032 and pSB17042 from poultry samples, and pHBI02, pHBI03, pHBI04, and pHBI05 from clinical isolates. Plasmids present in all ST429 and ST162 isolates in this study contained the additional resistant genes *sul1*, *aac (3)-Via*, *aadA1*, and *tet(A)*, except for plasmid pPS15001 (*tetA* was absent). We also found that six IncK plasmids from ST1286 isolates carried *bla*_TEM-1_ and *bla*_CMY-2_, which co-occurred with the *dfrA1*, *sul2*, and *tet(A)* resistant genes. VirulenceFinder detected *traT* (*n* = 28, 100%) and *cib* (*n* = 22, 78.6%) in all or most plasmids.

### 2.7. Comparison of IncK-bla_CMY-2_ Plasmids from This Study and Public Databases

A comparison of the IncK-*bla*_CMY-2_ plasmids under study showed that all have the conjugative transfer genes encoding pilus (*pilI*, *pilP*, *pilR*, *pilS*, *pilM*, *pilN*, and *pilV*), genes from the transference operon *tra* (*traC-F*, *traH*, *traJ*, *traM-T*, *traV-W*, and *traX-Y*), and genes encoding for a DNA primase (T7 DNA primase) and an endonuclease relaxase (*nikB*) (Figure 5 and Appendix A). They also shared genes involved in plasmid partition/stability. Interestingly, four plasmids of poultry origin belonging to ST429 (*n* = 3) and ST162 (*n* = 1), namely, pSB17040, pRS184-R, pRL97, pSA17101, and the plasmid pHBI01 from a human blood ST429 isolate, showed 99.9% identity: they displayed common virulence factors and resistance genes (Table 2) although their origin (production systems, slaughterhouses, or clinical, as well as years and countries of isolation) differed. The plasmid pPS15001, found in an *E. coli* ST429, lacked a 10 kb region with *tet(A)* and mercury-resistance genes compared to the remaining plasmids linked to ST429 (Table 2). Importantly, ST1286 *E. coli* isolates emerging in 2018 contained the largest IncK-*bla*_CMY-2_ plasmid (132–133 kb) that harbored some unique genes encoding a transposase, the toxin–antitoxin system (*relE*/*parE*), and the SOS inhibition genes *psiAB* not found in the other IncK-*bla*_CMY-2_ plasmids. However, the mercury-resistance operon (*mer*) was not detected in IncK-*bla*_CMY-2_ plasmids from *E. coli* ST1286 (Appendix A). IncK-*bla*_CMY-2_ plasmids from ST429 *E. coli* included in this study exhibited a high similarity with publicly available IncK plasmids present in *E. coli* ST429 from Norway and the USA (Appendix A), while a Norwegian IncK-*bla*_CMY-2_ plasmid from an *E. coli* ST162 isolate lacked a 45 kb fragment compared to the Danish pSA17101 from *E. coli* ST162. In addition, plasmids pSB17032 and pSB17042, from *E. coli* ST57 and ST350 isolated from slaughterhouses, were almost identical to the pHBI003 plasmid from the Danish human blood *E. coli* ST131 isolate, as shown in Figure 5.

The phylogenetic tree with 53 IncK-*bla*_CMY-2_ complete sequences from the GenBank database (Appendix A) and 28 IncK-*bla*_CMY-2_ plasmids from the current study is shown in Figure 6. Among these plasmids, 75 plasmids, showing a worldwide distribution, harbored only one β-lactamase gene, *bla*_CMY-2_, while six plasmids, restricted to Denmark, also contained the *bla*_TEM-1_ resistance gene. The plasmid sizes ranged from 70 kb to 133 kb. The conjugation-associated *traT* gene was found in 79 (98%) plasmids, while 40 plasmids showed, in addition, the virulence factor colicin Ib (*cib* gene), which is a polypeptide toxin acting against *E. coli* and closely related bacteria. Phylogenetic analysis showed that three plasmids, pPS15001 from Denmark, p30P2 from the USA (accession no. LC557961.1), and p22C121-2 from Japan (accession no. LC501554.1), containing the same additional antimicrobial-resistance (*aac (3)-Via, aadA1, sul1*) and virulence genes (*cib, traT*) clustered with plasmids that, besides the above-mentioned genes, also carried *tet(A)*, indicating a putative common ancestor and subsequent introduction of the *tet(A)* gene. The IncK-*bla*_CMY-2_ plasmid pN16S065 (accession no. CP082750.1) recovered from a *Salmonella enterica* isolate in the USA clustered with plasmids from *E. coli*, suggesting the spread of these plasmids to other Enterobacteriaceae. Moreover, IncK-*bla*_CMY-2_ plasmids isolated from human clinical cases were interspersed with plasmids from animals, food, and the environment, suggesting that IncK-*bla*_CMY-2_ plasmids and their hosts circulate in the entire ecosystem.

## 3. Discussion

This study characterized the genetic diversity of 46 IncK-*bla*_CMY-2_-positive ESC-resistant *E. coli* isolates previously collected from the Danish poultry production system and human clinical samples during 2015–2021, Finnish broilers in 2017, and German chicken meat in 2012. Using public databases and data from the present study, the epidemiology of ST429 *E. coli* isolates and IncK-*bla*_CMY-2_ plasmids in the context of poultry production and possible links to clinical cases was evaluated. The results indicated that genetically diverse ESC-resistant *E. coli* STs with IncK-*bla*_CMY-2_ plasmids have been circulating in the poultry production chain in Denmark and other countries for over a decade. Moreover, this study strongly suggests that transfers to humans are occurring with links to blood infections. Previous studies have also documented the spread of IncK-*bla*_CMY-2_ plasmids in ESC-resistant *E. coli* isolates obtained from 2006 to 2012 across different host species, including humans in Denmark [23]. 

Antimicrobial susceptibility testing showed resistance of all (*n* = 46) ESC-resistant *E. coli* isolates to ampicillin, cefotaxime, and ceftazidime. This resistance pattern resembled previous reports for veterinary isolates in other countries [14,24,25]. Moreover, the majority of the IncK-*bla*_CMY-2_-positive ESC-resistant *E. coli* isolates (43/46, 93.5%) presented a multidrug-resistance profile, i.e., being resistant to at least three classes of antimicrobials, which is higher than the 84% observed in a Spanish study of ESC-resistant *E. coli* isolates from a laying hen farm [26]. The most common plasmid-mediated co-resistance, found in the present study, to sulfamethoxazole (93.5%, 43/46), tetracycline (84.8%, 39/46), gentamicin (71.7%, 33/46), and trimethoprim (19.6%, 9/46), suggests a concerning spread of resistance among the IncK-*bla*_CMY-2_-positive ESC-resistant *E. coli* population, especially since tetracycline, penicillin, and sulfamethoxazole, in combination with trimethoprim, were reported to be the most commonly used antimicrobials for poultry farming in Denmark in 2016 [27]. Interestingly, resistance levels of 21.3, 25.9, 69.7, and 22.8% against cefotaxime, tetracycline, sulfamethoxazole, and trimethoprim, respectively, were recently reported in *E. coli* isolated from cloacal swabs from Danish broiler flocks in 2016–2017 [28], pointing to the presence of multidrug-resistant *E. coli* strains in the poultry production pyramid. 

All the *bla*_CMY-2_-positive *E. coli* isolates carry multiple resistance genes, with the in silico results agreeing with the phenotypic analysis (Figure 2, Table 1). The gentamicin-resistance genes *aac (3)-Vla* and *aac (3)-IId* were frequently detected in the IncK-*bla*_CMY-2_-positive ESC-resistant *E. coli* isolates, genes also observed in *E. coli* from chicken and human sources in Canada [29]. 

One human isolate (HBI02) harbored both the azithromycin-resistance gene *mph(A)* and the chloramphenicol-resistance gene *catA1*. Co-harborage of the *bla*_CMY-2_, *mph(A)*, and *catA1* genes was previously reported in *E. coli* isolates from calves in the USA [30]. A recent study from Korea detected the *catA1* gene in *bla*_CMY-2_-producing pathogenic *E. coli* in pigs, but there were no chloramphenicol-resistant genes in the strains isolated from humans [31]. In addition, four of the IncK-*bla*_CMY-2_-positive *bla*_CMY-2_-positive *E. coli* isolates showed chromosomal point mutations yielding fluoroquinolone resistance (Figure 1). These isolates had at least two mutations in the *gyrA* gene combined with high MIC values of ciprofloxacin (0.25–0.5 mg/L) and simultaneously nalidixic acid MIC values ≥ 128 mg/L. Single mutations in the *gyrA* gene altering leucine at position 83 (S83L), as well as double mutations in the same gene altering positions at aspartic acid (D87N) and serine (S83L), are known to turn *E. coli* strains resistant to fluoroquinolones [32]. 

All 46 IncK-*bla*_CMY-2_-positive ESC-resistant *E. coli* could transfer the *bla*_CMY-2_ gene by conjugation to a recipient, pointing to the potential for horizontal transmission of the plasmid-borne *bla*_CMY-2_ gene. A previous study in Denmark identified that exogenous *E. coli* of human or animal origin could readily transfer *bla*_CMY-2_-encoding plasmids to the human fecal microbiota [33]. Furthermore, an IncK-*bla*_CMY-2_ plasmid was transferable between *E. coli* and *S. Heidelberg* isolates, but the transfer was unsuccessful between *S. Heidelberg* isolates, as described by [34]. Also, a previous study demonstrated the presence of IncK-*bla*_CMY-2_ plasmids in *Salmonella enterica* in the USA [35], plasmids which interestingly were similar to the ones detected in the present study (Figure 6). 

The pangenome analysis revealed that the IncK-*bla*_CMY-2_ *E. coli* isolates possess a large source gene pool and the capacity to acquire novel genetic elements. The pool of conserved core genes is three times smaller than the pools of accessory and cloud genes, which suggests a flexible genome [36]. Other studies have also described the core genome of *E. coli* as being comparatively small [37,38,39]. However, it is worth emphasizing that the core genome is relative, as the concatenated core would become smaller if more genomes were added to the comparison [40]. On the other hand, it is notable that the smaller size of core genomes will result in more expansive accessory genomes and isolate-specific cloud genes [41]. Of course, these non-essential accessory and cloud genes are prone to rapid evolution, and their knockout does not impact the isolate phenotype [42]. In addition, the tree provides deeper insight into the concatenated core gene alignment, which indicates that various *E. coli* STs have been of great significance in the evolution of IncK-*bla*_CMY-2_ *E. coli* isolates, as previously described [43].

IncK-*bla*_CMY-2_-positive ST429 *E. coli* isolates with 0–54 SNP differences were originally isolated from various stages of poultry production and from humans, implying that clonal transmission happens between different hosts. ST429 isolates from Finland and Germany included in this study were previously demonstrated to show few SNP differences in each case/country [14,22]. In addition, ST1286 isolates with 13–90 SNP differences were originally collected from slaughterhouses [20], also suggesting clonal transmission. A recent study from China reported no SNP differences among ST1286 *E. coli* isolates from laying hens [44]. While ST1286 IncK-*bla*_CMY-2_-positive *E. coli* isolates were found in poultry, this ST was not among our IncK-*bla*_CMY-2_-positive ESC-resistant *E. coli* isolates from humans, indicating lower virulence potential, in agreement with previous studies [45,46]. 

The cgMLST analysis of ST429 *E. coli* isolates from Denmark [20] and Enterobase (*n* = 81) showed that the isolates differed in a limited number of alleles, highlighting the existence of a conserved pool of ST429 carrying *bla*_CMY-2_ and supporting the transmission of *bla*_CMY-2_ along the poultry production chain and across sectors. Moreover, the cgMLST analysis of ST429 *E. coli* isolates carrying other *bla*-family genes such as *bla*_CTX-M_ also identified the clonal relationship between isolates from different countries [47]. In addition, a recent pan-European study reported ESC-resistant *E. coli* isolates carrying *bla*_CMY-2_ on different Inc plasmid types (e.g., IncI1, IncK2, IncA/C) in diverse STs, reflecting the dissemination of cephalosporin-resistance genes via successful plasmid lineages [48]. 

IncK plasmids of different sizes and genetic contents were demonstrated by short-read and long-read sequencing to contain a conserved carrier/location of the *bla*_CMY-2_ gene and to occur in genetically diverse STs. IncK plasmids can be divided into two separate lineages, namely, IncK1, which is often associated with *bla*_CTX-M-14_, and IncK2, which predominately carries *bla*_CMY-2_ [49,50]. Previous studies have described, using hybrid assembly, transferable IncK-*bla*_CMY-2_ plasmids in ESC-resistant *E. coli* isolates in other countries [51,52]. Here, we found that one ST429 Danish *E. coli* harbored a smaller (~109 kb) IncK-*bla*_CMY-2_ plasmid, while the remaining eight ST429 isolates from Denmark harbored a longer version (119–120 kb). Thus, our results suggest that at least two variants of IncK-*bla*_CMY-2_ plasmids are circulating among ST429 ESC-resistant *E. coli* isolates in the poultry production chain in Denmark. To note, the most common Danish IncK-*bla*_CMY-2_ plasmids were highly similar to plasmids originating from broiler production in other countries, IncK-*bla*_CMY-2_ plasmids circulating in Germany (117–120 kb) in 2012, in Finland (~122 kb) in 2017 [14,43], and in Norway (~110 kb) [53], indicating the successful spread of IncK-*bla*_CMY-2_ plasmids in broiler production. Also, highly similar IncK-*bla*_CMY-2_ plasmids linked to three different *E. coli* STs from two slaughterhouses were observed, indicating a potential horizontal dissemination among *E. coli* STs in broiler production, as suggested by others [54]. 

Among the five IncK-*bla*_CMY-2_-positive *E. coli* clinical isolates linked to different STs, one clinical isolate of *E. coli* ST429 harbored a IncK-*bla*_CMY-2_ plasmid (pHBI01), which shared high sequence homology with those from *E. coli* ST429 from poultry (Figure 5), providing evidence of the vertical clonal spread of *E. coli* harboring IncK-*bla*_CMY-2_ plasmids between poultry and humans. While *E. coli* ST429 isolates represent a common avian pathogenic lineage specific to poultry, it was previously believed to hold little pathogenic potential for humans [55]. However, extraction of *E. coli* ST429 isolates from Enterobase revealed the presence of eight isolates harboring IncK-*bla*_CMY-2_ plasmids, which were derived from human clinical cases during 2016–2021 (Appendix A). Taken together, this may imply a stronger pathogenicity potential for this ST than previously thought. 

To note, the IncK-*bla*_CMY-2_ plasmids pHBI02 and pHBI03, linked to *E. coli* ST69 and ST131 isolates from humans, exhibit a high similarity with IncK-*bla*_CMY-2_ plasmids related to *E. coli* ST350 and ST57 from slaughterhouse meat (Figure 5), indicating a horizontal transfer of plasmids from poultry to clinical *E. coli* lineages in Denmark. *E. coli* ST131 is responsible for 50% of ESBL blood infections with no recognized animal reservoir [56], although a link to broilers was implied in a 2019 study [19]. *E. coli* ST69 is a globally distributed *E. coli* responsible for hospital-acquired antimicrobial-resistant human infections [57] and known to be able to colonize the animal intestine [58]. *E. coli* ST1286 (*n* = 6 strains), according to DANMAP [59], is not a frequent ST associated with *bla*_CMY-2_ in broilers or broiler meat, which agrees with our previous study [20], where ST1286 isolates carrying *bla*_CMY-2_ were only isolated from poultry meat from one slaughterhouse in 2017–2018. 

The detailed plasmid comparison performed in this study revealed a common IncK-*bla*_CMY-2_ backbone sequence of 85 kb for all 28 plasmids. This plasmid backbone sequence might be common in poultry, as described earlier [50,54]. Additionally, most of our IncK-*bla*_CMY-2_ plasmids contain toxin–antitoxin systems (e.g., pndC/ydfB or relE/ParE) and regions involved in plasmid transfer (such as *tra*, *pil* operons) that ensure stable maintenance of the plasmid to the daughter cells after cell division and the spread to other strains, respectively, which increases the potential for transmission to opportunistic pathogens in the poultry and human gut microbiota or in environmental reservoirs [60]. Interestingly, a mercury-resistance operon (*mer*, [61]) was observed in IncK-*bla*_CMY-2_ plasmids from *E. coli* ST429 originally isolated from poultry and clinical samples in Germany, Finland, and Denmark [14,20,43], whereas *mer* genes were not detected in IncK-*bla*_CMY-2_-positive *E. coli* from other STs (poultry isolates, ST1286, ST162, ST350, and ST57; and clinical isolates, ST69, ST131, ST95, and ST12). It is possible that contamination of poultry or other animal feed in the farms promoted the carriage of mercury genes in IncK-*bla*_CMY-2_ plasmids. Along these lines, mercury was detected in mineral feed used in poultry rearing in Germany in 2013 [62]. 

In this study, we created a phylogenetic tree of IncK-*bla*_CMY-2_ plasmids composed of sequences from this study (*n* = 28) and publicly available sequences at GenBank, NCBI (*n* = 53, from seven other countries). To our knowledge, this is the first attempt to compare all IncK-*bla*_CMY-2_ plasmids published so far (*n* = 81). Results highlight that IncK plasmids represent a major vehicle for *bla*_CMY-2_ and other antimicrobial-resistance genes worldwide and over time, overall, across the poultry production chain but also in humans. *E. coli* was the dominant host with only one IncK-*bla*_CMY-2_ plasmid carried by *Salmonella enterica* [63]. Recent studies have reported that *bla*_CMY-2_ is also harbored by IncI1 plasmids present in *Salmonella enterica* isolates from chicken meat in Spain and South Korea [64,65]. Since 39 (48.1%) IncK-*bla*_CMY-2_ plasmids were confirmed to carry genes encoding resistance to sulfonamides (*n* = 35) and tetracyclines (*n* = 25) all over the world, the spread of such plasmids might be prompted by the use of these antimicrobial agents worldwide. A total of 79 (97.5%) IncK-*bla*_CMY-2_ plasmids harbored the *traT* gene and 40 (49.4%) contained the *cib* genes which indicates the horizontal transfer of the IncK-*bla*_CMY-2_ plasmid and co-location of *bla*_CMY-2_ with virulence factor colicin Ib, respectively [66]. A previous study showed that the mutation of the *traY* gene of IncK plasmids effectively prevents conjugation [67]. 

## 4. Materials and Methods

### 4.1. IncK-bla_CMY-2_-Positive ESC-Resistant E. coli Isolates Included in This Study

A total of 46 ESC-resistant *E. coli* isolates from different STs were selected from four unrelated collections [20,22] based on their carriage of IncK-*bla*_CMY-2_ plasmids and association with poultry production or human clinical cases. The selection consisted of 31 (linked to ST429, ST1286, ST162, ST350, or ST57) isolates obtained from the poultry production system and slaughterhouses in Denmark, collected over the period 2015–2018 [20], seven ST429 *E. coli* isolates obtained from Finnish poultry production in 2017 [22], three *E. coli* ST429 isolates originating from German chicken meat in 2012 [14], and five isolates (linked to ST429, ST69, ST131, ST95, ST12) originating from Danish clinical bloodstream infections during 2017–2021. More detailed information for all isolates is shown in Table 2.The isolates were routinely cultivated on MacConkey agar (Oxoid, Basingstoke, UK) supplemented with 1 mg/L cefotaxime (Sigma-Aldrich, St. Louis, MO, USA) followed by transfer of colony mass into 1 mL of LB broth with 18% glycerol for long-term storage at −80 °C. Genomes from all the isolates were previously sequenced using different Illumina platforms ([14,20,22], and H. Hasman personal communication [68]). The raw data were quality trimmed (Q20) using Trimmomatic (v.0.36) [69] and assembled using SPAdes software (v.3.11.1) [70]. 

### 4.2. Antimicrobial Susceptibility Testing

The antimicrobial susceptibility profiles of the 46 *E. coli* isolates were evaluated by determining the Minimum Inhibitory Concentration (MICs) using Sensititre^®^ EUVSEC3^®^ plates (Thermofisher Scientific, Paisley, UK) following the manufacturer’s recommended protocol. The EUVSEC3^®^ plate contains 15 antibiotic agents of significance to public health, including the tested range of amikacin (4–128 μg/mL), ampicillin (1–32 μg/mL), azithromycin (2–64 μg/mL), cefotaxime (0.25–4 μg/mL), ceftazidime (0.25–8 μg/mL), chloramphenicol (8–64 μg/mL), ciprofloxacin (0.015–8 μg/mL), colistin (1–16 μg/mL), gentamicin (0.5–16 μg/mL), meropenem (0.03–16 μg/mL), nalidixic acid (4–64 μg/mL), sulfamethoxazole (8–512 μg/mL), tetracycline (2–32 μg/mL), tigecycline (0.25–8 μg/mL), trimethoprim (0.25–16 μg/mL). Results were interpreted based on the Epidemiological Cut-Off value (ECOFF) issued by the European Committee on Antimicrobial Susceptibility Testing (EUCAST), except for sulfamethoxazole, for which the sensibility or resistance values were reported according to the EU surveillance ECOFF [71,72]. The MICs for *E. coli* ATCC^®^ 25922 were also tested for quality control. Three technical replicates were performed for each isolate. 

### 4.3. Plasmid Conjugation

Transfer of the plasmids was confirmed in a filter conjugation mating assay with some modifications [73]. All *bla*_CMY-2_ carrying ESC-resistant *E. coli* isolates were used as donors, and *E. coli* MG1655 resistant to rifampicin and nalidixic acid served as a recipient strain. Donor and recipient strains were cultured overnight in Luria-Bertani broth (LB, Sigma-Aldrich, St. Louis, MO, USA) at 37 °C and washed twice with phosphate-buffed saline (PBS, Invitrogen, Maryland, MD, USA). After adjusting the OD600 to 0.5, donor and recipient isolates were mixed at a ratio of 1:1, and 100 μL were immediately applied to a 0.2 μm nitrocellulose filter membrane, placed on LB agar, followed by overnight culture at 37 °C for 20 h. Subsequently, serial decimal dilutions of the cultures embedded in the filters were prepared in sterile saline solution, and transconjugants were selected by cultivation on LB agar containing appropriate antibiotics: cefotaxime (2 μg/mL); or rifampicin (100 μg/mL) and nalidixic acid (100 μg/mL) (Sigma-Aldrich, St. Louis, MO, USA). Donor and recipient strains were spread on LB agar supplement with cefotaxime, or with rifampicin and nalidixic acid, respectively, used as controls. All presumed transconjugants were confirmed to contain *bla*_CMY-2_ by PCR using previously described primers and conditions [74]. Each experiment consisted of three biological replicates and three technical replicates. 

### 4.4. Genome Annotation and Pangenome Analysis

All de novo assemblies of the isolates’ genomes were carried out using the SPAdes (v.3.11.1), and the annotation was performed using the BAKTA annotation pipeline v1.7.0 with the default setting [75]. Core and accessory genome comparison analyses of the ESC-resistant *E. coli* isolates were performed using the Roary pangenome pipeline v3.13.0 [76]. Roary produces a core gene alignment result from gff3 files created by BAKTA annotation. The “core” genes were identified in the isolates using a 99% identity cut-off. A concatenated core gene alignment of all isolates’ core genes was generated and attached as a Appendix A. The combined core gene alignment was used to construct an approximately maximum-likelihood phylogenetic tree using SNP sites [77] and the tree with FastTree v2.1 [78]. The gene presence/absence file obtained by the Roary pangenome annotation pipeline and the core gene phylogenetic tree were visualized using a web-based interactive visualization of the genome tool Phandango [79].

### 4.5. In Silico Analysis for Sequence Type and Resistance Genes

Sequencing data from the isolates, except for the isolates reported in the previous study [20], were analyzed using the web-based Center for Genomic Epidemiology (CGE) tools (https://cge.cbs.dtu.dk/services/, accessed on 15 May 2023). STs and resistance genes or chromosomal point mutations yielding resistance to specific antibiotics were confirmed using MLST 2.0 [80] and ResFinder 4.1 [81], respectively. 

### 4.6. Isolate Phylogenetic Analysis

To detect/compare SNP (single nucleotide polymorphism) differences among ST429 and ST1286 isolates, CSIPhylogeny (https://cge.dtu.dk/services/CSIPhylogeny/, accessed on 10 June 2023) was used on the CGE server with default settings and ignoring heterozygous SNPs. ST429 and ST1286 input sequences were mapped to the earliest ESC-resistant *E. coli* genomes, i.e., PS15001 and SB17103 for ST429 and ST1286, respectively, as the reference strains to call SNPs and searched for the previously described nucleotide variations [82]. The following criteria for high-quality SNP calling and filtering were chosen: (i) select a minimum depth of 10× at SNP positions; (ii) select the minimum relative depth of 10% at SNP positions; (iii) select a minimum distance of 10 bp between SNPs; (iv) select minimum SNP quality of 30; (v) select minimum read mapping quality of 25; and (vi) select a minimum Z-score of 1.96. Site validation for each SNP position was performed. For bootstrap, 1000 replicates were generated to construct the tree. Two SNP matrices were created in MS Excel for each pair of strains, and their phylogeny trees were visualized using iTOL (http://itol.embl.de/, accessed on 15 June 2023) [83]. 

To further compare the relationship among ST429 isolates worldwide, a search for *E. coli* ST429 by the Achtman 7-gene MLST in EnteroBase [84] provided 371 results up to 15 June 2023 (http://enterobase.warwick.ac.uk, accessed on 15 March 2023). Of these, 80 came with relevant metadata to be considered for further analysis and are provided in a Appendix A. The metadata of interest were the source, sample name, year of isolation, and country of origin. To compare this study’s 22 ST429 Danish poultry isolates and the 80 Enterobase ST429 isolates, a phylogenetic analysis using an ad hoc core genome multilocus sequence typing (cgMLST) scheme with 2513 genes [20] was performed using Ridom Seqshere+ (v5.0.1, Ridom GmbH, Munster, Germany) [85] and visualized with a minimum spanning tree. 

### 4.7. Plasmid Sequencing

To obtain high-quality IncK-*bla*_CMY-2_ plasmid sequences, 28 out of the 46 ESC-resistant *E. coli* isolates were selected to represent different STs, sources, and countries, and subjected to long-read sequencing (Table 2). All isolates were grown on MacConkey agar (Oxoid, Basingstoke, UK) with 1 mg/L cefotaxime (Sigma-Aldrich, St. Louis, MO, USA) overnight at 37 °C and DNA was then extracted using Genomic-Tip G/500 kit (Qiagen, Hilden, Germany) following the manufacturer’s protocol. Libraries were constructed with the 1 D Ligation Barcoding Kit (catalog no. SQK-RBK114.96, ONT, Oxford, UK) according to the manufacturer’s protocol and were sequenced on a R10.4.1 flowcell (FLO-MIN114) used with a MinION Mk1B sequencing device and sequenced with MinKNOW software v4.5.5 for 20–24 h. Long reads in the fast5 format were base called, demultiplexed, and converted into fastq format using Guppy v6.4.4 (ONT). The adaptor sequences were removed using Porechop v0.2.2 [86]. Hybrid assembly of long and short reads using Unicycler v0.4.0 [87] resulted in circular contigs of the plasmid. We mapped the short Illumina reads to the plasmid contigs and performed error correction using CLC Genomic Workbench v.11.0.1 (QIAGEN, Aarhus, Denmark) by calling variants based on the mapping. Manual correction of errors was needed since homopolymer areas in sequences were especially problematic for the MinION technology. 

### 4.8. Plasmid Comparison

The 28 assembled IncK-*bla*_CMY-2_ plasmid sequences were annotated using BAKTA annotation v1.7.0 [75]. Three additional published IncK-*bla*_CMY-2_ plasmids (2016-40-16449, p2016-40-16852, and p2) found in ESC-resistant *E. coli* isolated in Norwegian and American poultry production [53,88] were included in the comparison. Resistance and virulence genes and plasmid types were determined using ResFinder 4.1 [81], VirulenceFinder 2.0 [89], and PlasmidFinder 2.1 [90], respectively. Alignment of plasmid sequences with similar structures was generated by BLAST Ring Image Generator (BRIG) v0.95 analysis [91] and the clinker webserver (https://cagecat.bioinformatics.nl, accessed on 15 May 2023) [92]. 

### 4.9. Plasmid Phylogenetic Tree

Complete sequences of IncK-*bla*_CMY-2_ plasmids were recovered from the GenBank databases using a keyword search for the words “IncK” and “*bla*_CMY-2_”. All 81 sequenced IncK-*bla*_CMY-2_ plasmids from the current study (*n* = 28) and GenBank (*n* = 53) were annotated by BAKTA [75]. Moreover, antimicrobial-resistance genes and plasmid types were confirmed using the CGE tools as described above, and plasmids were analyzed for the content of virulence genes (virulence factor databases (VFDB) [93] and VirulenceFinder. Subsequently, SNP sites generate SNP alignments [77]. An approximate maximum-likelihood phylogenetic tree of IncK-*bla*_CMY-2_ plasmids was constructed with FastTree v2.1 under the general-time reversible model with a categorical model of the rate heterogeneity (GTR-CAT), based on the Roary method of alignment [78]. The phylogenetic tree was visualized using iTOL tool v4.3.3 (http://itol.embl.de/, accessed on 15 May 2023) [83].

## 5. Conclusions

In this study, we investigated the persistence and dynamics of 46 ESC-resistant *E. coli* and their IncK-*bla*_CMY-2_ plasmids, previously isolated from poultry and humans. Our results revealed that different *E. coli* STs carried highly similar IncK-*bla*_CMY-2_ plasmids, with the most common ST429 being isolated both from poultry and a human blood infection. The presence of highly similar plasmids in different *E. coli* STs could be due to the persistence of IncK-*bla*_CMY-2_. Furthermore, ST429 *E. coli bla*_CMY-2_ isolates were found to occur globally pointing toward a common ancestor that has spread between the various reservoirs. The distribution of IncK-*bla*_CMY-2_ plasmids also provided evidence for the worldwide spread of IncK-*bla*_CMY-2_ producing ESC-resistant *E. coli* isolates. Further surveillance of IncK-*bla*_CMY-2_ plasmids in different *E. coli* STs in poultry production chain and humans should be carried out in order to increase our understanding of the dynamics of these ESC-resistant *E. coli*.

## Figures and Tables

**Figure 1 antibiotics-13-00349-f001:**
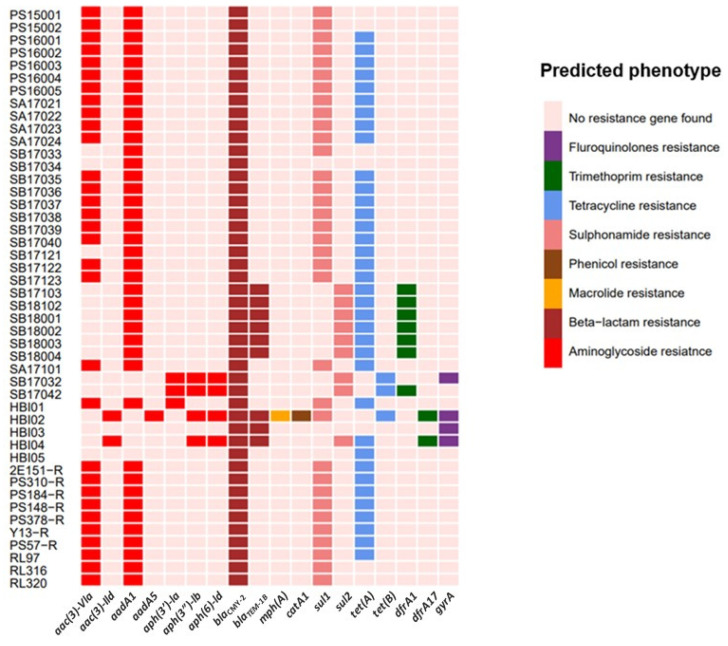
Antimicrobial-resistance gene determinants were identified in the IncK-*bla*_CMY-2_-positive ESC-resistant *E. coli* (*n* = 46) using Resfinder 2.0 on the CGE tool. The heatmap shows the presence or absence of antimicrobial-resistance genes in each isolate. Rows and columns represent isolates and predicted antimicrobial-resistance genes, respectively. Colors indicate the predicted resistance phenotype to different classes of antibiotics for each isolate based on genotype.

**Figure 2 antibiotics-13-00349-f002:**
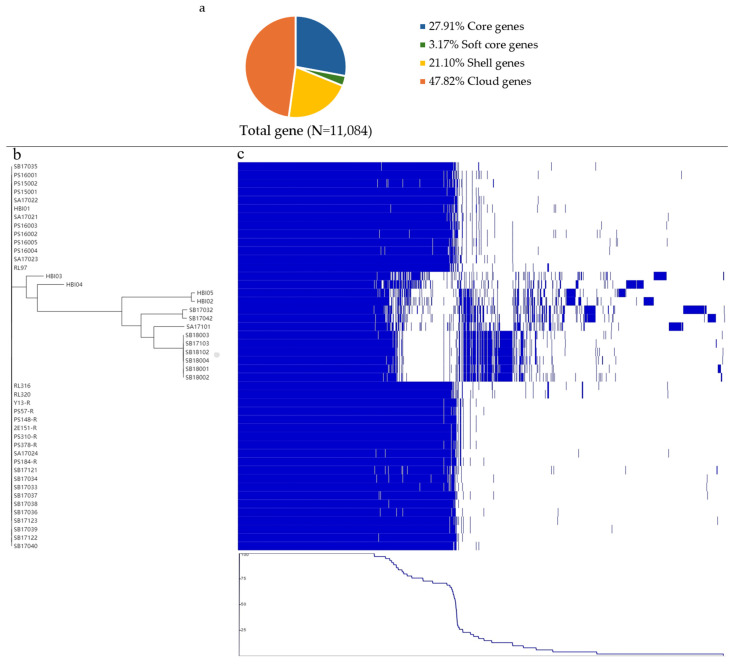
Pangenome analysis of all ESC-resistant *E. coli* carrying IncK-*bla*_CMY-2_ (*n* = 46). (**a**) Distribution of total genes (100%): core genes (27.91%) found in ≥99%, soft-core genes (3.17%) found in between 95% and 99%, shell genes (21.10%) found in between 15% and 95%, and cloud genes found in <15% of the *E. coli* in the study. (**b**) Maximum likelihood phylogenetic tree inferred from the alignment of the 3094 core genes of 46 ESC-resistant *E. coli* by FastTree. (**c**) Annotation of gene presence (blue) and absence (white) matrix across the pangenome of the *E. coli*. The top scale shows the complete genome size (kbp). Each row shows the gene content of an *E. coli* isolate. Each column shows the comparative gene clusters. The data were visualized using Phandango.

**Figure 3 antibiotics-13-00349-f003:**
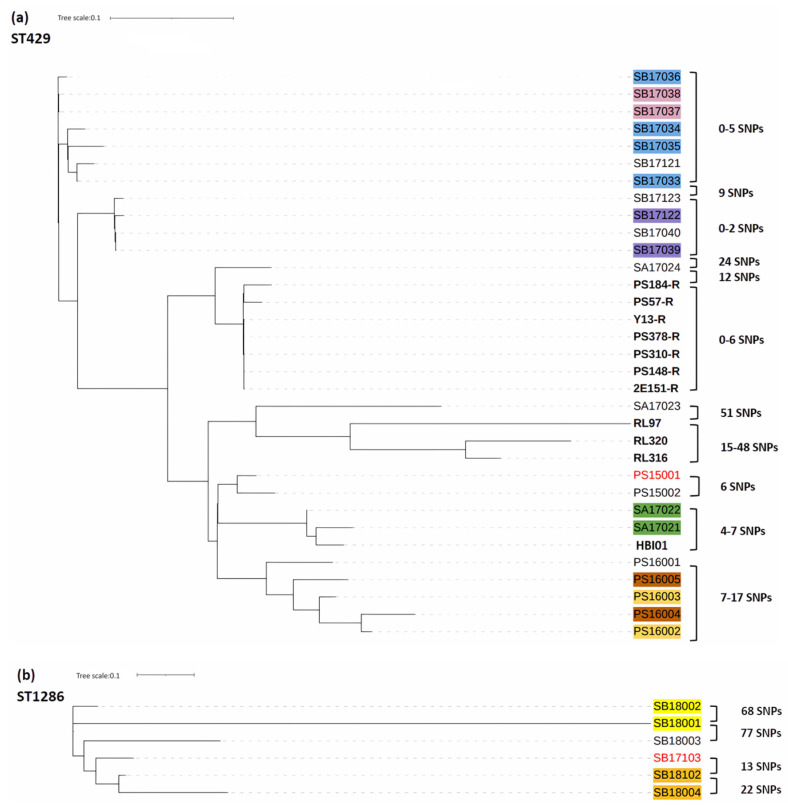
Phylogenetic trees show the relationships between ST429 (**a**) and ST1286 (**b**) ESC-resistant *E. coli* isolates collected from Danish, Finnish, and German poultry and one Danish human sample (see Table 2 for isolate information). Each SNP-based tree was constructed with CSI Phylogeny 1.4 (https://cge.dtu.dk/services/CSIPhylogeny/, accessed on 15 May 2023), using the genomes of PS15001 and SB17103 (in red) as a reference in each case. For ST429, poultry isolates from Finland and Germany and 1 human isolate from Denmark are shown in black bold. Isolates from the same date and source are presented in the same color. The observed number of SNPs among these isolates is indicated as well.

**Figure 4 antibiotics-13-00349-f004:**
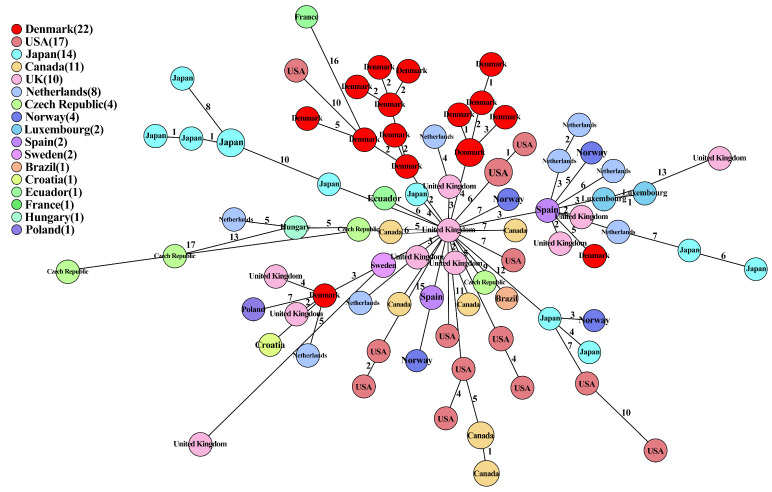
Minimum spanning tree of 102 ST429 *E. coli* isolates collected from poultry production (this study, Danish isolates, *n* = 22) and Enterobase (*n* = 80) based on an ad hoc 2513 gene cgMLST scheme and 7 *E. coli* MLST Warwick targets calculated in Ridom Seqshere+. Isolates are colored according to country and year of isolation indicated in circles (each circle may contain more than one isolate in cases of no allelic differences). The number of isolates per country is indicated in brackets in the color legend. Numbers of allelic differences between isolates are shown in the connecting lines. More detailed information on isolates and the minimum spanning trees can be found in Appendix A.

**Figure 5 antibiotics-13-00349-f005:**
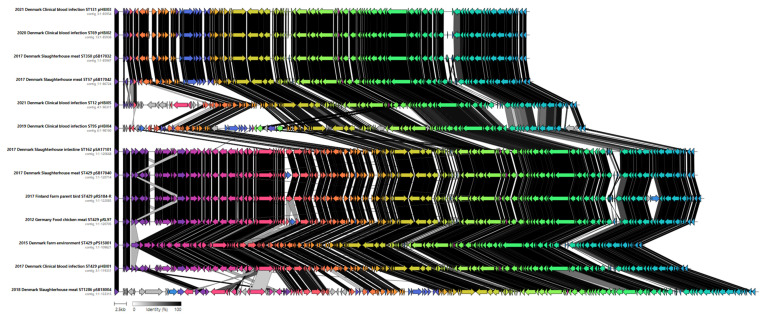
Overall linear comparison analysis of IncK-*bla*_CMY-2_ plasmids (*n* = 13) linked to different STs (ST131, ST69, ST350, ST57, ST12, ST95, ST162, ST429, ST1286).

**Figure 6 antibiotics-13-00349-f006:**
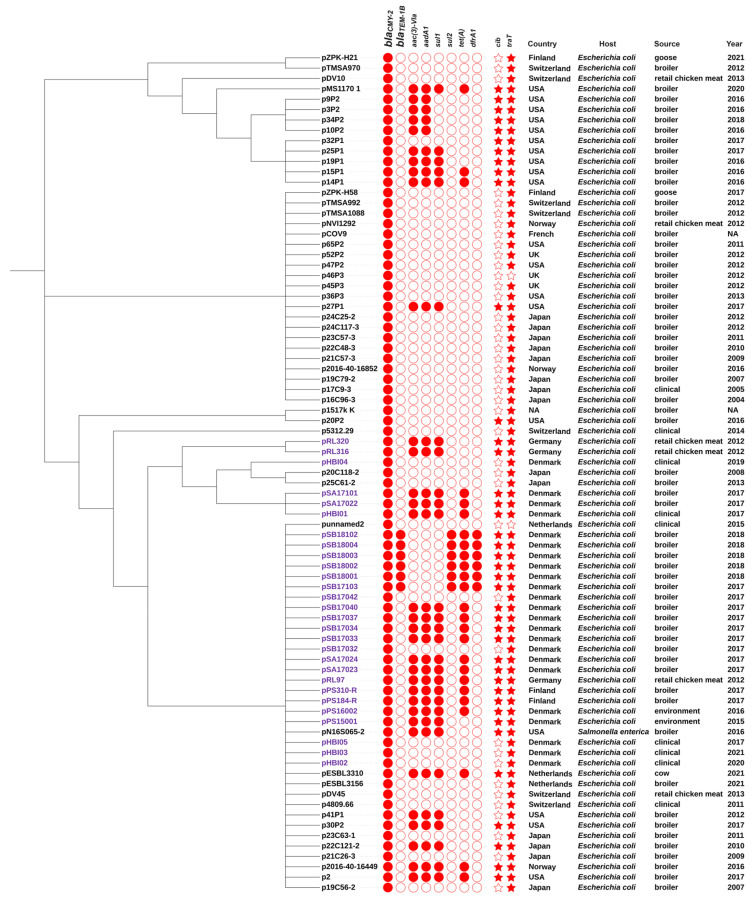
FastTree approximate maximum-likelihood phylogenetic tree of IncK-*bla*_CMY-2_ plasmids constructed with plasmids from this study (*n* = 28) labeled in purple and from GenBank, NCBI, labeled in black font (*n* = 53). The heatmap denotes the presence/absence of antimicrobial-resistance and virulence genes on the plasmids. The following annotation from left to right represents country, species, source, and year of isolation.

**Table 1 antibiotics-13-00349-t001:** Distribution of MICs of several antibiotics in the ESC-resistant *E. coli* (*n* = 46) isolates included in the study.

	Resistant Strains (%)	Distribution (%) of Strains Linked to a Specific MIC Value (mg/L) *															
Substance		0.015	0.03	0.06	0.12	0.25	0.5	1	2	4	8	16	32	64	128	256	≥512
Amikacin	0									100							
Ampicillin	100												2.2	97.8			
Azithromycin	2.2								13.0	78.3	6.5			2.2			
Cefotaxime	100								2.2	8.7	89.1						
Ceftazidime	100								2.2	2.2	45.6	50.0					
Chloramphenicol	2.2										93.5	4.3	2.2				
Ciprofloxacin	8.6	69.6	21.8			4.3	4.3										
Colistin	0							100									
Gentamicin	71.7						8.7	17.4	2.2			23.9	47.8				
Meropenem	0		100														
Nalidixic acid	8.7									91.3					8.7		
Sulfamethoxazole	93.5											6.5					93.5
Tetracycline	84.8								15.2					84.8			
Tigecycline	0					97.8	2.2										
Trimethoprim	19.6					52.2	23.9	4.3					19.6				

* Black vertical lines represent epidemiological cut-off values (ECOFFs) for resistance. Light blue fields represent the range of dilutions tested for each antimicrobial agent, and light gray fields represent the non-tested dilutions. The % of isolates is for antimicrobial resistance within the range under study. When the growth of isolates was observed at the highest concentration of the antibiotic tested, the MIC value was recorded as the following concentration (e.g., for ampicillin, 97.8% of the isolates exhibited visible growth at a concentration of 32 mg/L, revealing an MIC of >32 mg/L and an annotation of the final MIC as 64 mg/L).

**Table 2 antibiotics-13-00349-t002:** Overview of characteristics of IncK-*bla*_CMY-2_ plasmids present in the 46 ESC-resistant *E. coli* isolated from poultry, meat, and clinical samples, collected in different countries over the period 2012–2020.

Isolate ID	Host Species	Sample Type	Year	Country *	ST			IncK Plasmid Characterization				
						ONT Sequenced	Plasmid Name	Plasmid Size (bp)	ESC-Resistance Gene	Other AMR Genes	Virulence Genes & Mercury-Resistance Genes	Conjugative Transferability of *bla*_CMY-2_
PS15001	Farm ^a^	Environment	2015	DK	429	+	pPS25001	109821bp	*bla* _CMY-2_	*sul1*, *aac (3)-Via*, *aadA1*	*cib*, *traT*	Positive
PS15002	Farm	Environment	2015	DK	429							Positive
PS16001	Farm	Environment	2016	DK	429							Positive
PS16002	Farm	Environment	2016	DK	429	+	pPS16002	119378bp	*bla* _CMY-2_	*sul1*, *aac (3)-Via*, *aadA1*, *tet(A)*	*cib*, *traT*, *mer*	Positive
PS16003	Farm	Environment	2016	DK	429							Positive
PS16004	Farm	Environment	2016	DK	429							Positive
PS16005	Farm	Environment	2016	DK	429							Positive
SA17021	Slaughterhouse ^b^	Thighs	2017	DK	429							Positive
SA17022	Slaughterhouse	Thighs	2017	DK	429	+	pSA17022	119378bp	*bla* _CMY-2_	*sul1*, *aac (3)-Via*, *aadA1*, *tet(A)*	*cib*, *traT*, *mer*	Positive
SA17023	Slaughterhouse	Thighs	2017	DK	429	+	pSA17023	119378bp	*bla* _CMY-2_	*sul1*, *aac (3)-Via*, *aadA1*, *tet(A)*	*cib*, *traT*, *mer*	Positive
SA17024	Slaughterhouse	Thighs	2017	DK	429	+	pSA17024	120714bp	*bla* _CMY-2_	*sul1*, *aac (3)-Via*, *aadA1*, *tet(A)*	*cib*, *traT*, *mer*	Positive
SB17033	Slaughterhouse	Thighs	2017	DK	429	+	pSB17033	120714bp	*bla* _CMY-2_	*sul1*, *aac (3)-Via*, *aadA1*, *tet(A)*	*cib*, *traT*, *mer*	Positive
SB17034	Slaughterhouse	Thighs	2017	DK	429	+	pSB17034	120714bp	*bla* _CMY-2_	*sul1*, *aac (3)-Via*, *aadA1*, *tet(A)*	*cib*, *traT*, *mer*	Positive
SB17035	Slaughterhouse	Thighs	2017	DK	429							Positive
SB17036	Slaughterhouse	Thighs	2017	DK	429							Positive
SB17037	Slaughterhouse	Thighs	2017	DK	429	+	pSB17037	120714bp	*bla* _CMY-2_	*sul1*, *aac (3)-Via*, *aadA1*, *tet(A)*	*cib*, *traT*, *mer*	Positive
SB17038	Slaughterhouse	Thighs	2017	DK	429							Positive
SB17039	Slaughterhouse	Thighs	2017	DK	429							Positive
SB17040	Slaughterhouse	Thighs	2017	DK	429	+	pSB17040	120714bp	*bla* _CMY-2_	*sul1*, *aac (3)-Via*, *aadA1*, *tet(A)*	*cib*, *traT*, *mer*	Positive
SB17121	Slaughterhouse	Intestine	2017	DK	429							Positive
SB17122	Slaughterhouse	Intestine	2017	DK	429							Positive
SB17123	Slaughterhouse	Intestine	2017	DK	429							Positive
SB17103	Slaughterhouse	Intestine	2017	DK	1286	+	pSB17103	132580bp	*bla*_CMY-2_, *bla*_TEM-1_	*dfrA1*, *sul2*, *tet(A)*	*cib*, *traT*	Positive
SB18102	Slaughterhouse	Intestine	2018	DK	1286	+	pSB18102	133311bp	*bla*_CMY-2_, *bla*_TEM-1_	*dfrA1*, *sul2*, *tet(A)*	*cib*, *traT*	Positive
SB18001	Slaughterhouse	Thighs	2018	DK	1286	+	pSB18001	132529bp	*bla*_CMY-2_, *bla*_TEM-1_	*dfrA1*, *sul2*, *tet(A)*	*cib*, *traT*	Positive
SB18002	Slaughterhouse	Thighs	2018	DK	1286	+	pSB18002	133297bp	*bla*_CMY-2_, *bla*_TEM-1_	*dfrA1*, *sul2*, *tet(A)*	*cib*, *traT*	Positive
SB18003	Slaughterhouse	Thighs	2018	DK	1286	+	pSB18003	133303bp	*bla*_CMY-2_, *bla*_TEM-1_	*dfrA1*, *sul2*, *tet(A)*	*cib*, *traT*	Positive
SB18004	Slaughterhouse	Thighs	2018	DK	1286	+	pSB18004	133315bp	*bla*_CMY-2_, *bla*_TEM-1_	*dfrA1*, *sul2*, *tet(A)*	*cib*, *traT*	Positive
SA17101	Slaughterhouse	Intestine	2017	DK	162	+	pSA17101	120668bp	*bla* _CMY-2_	*sul1, aac (3)-Via*, *aadA1*, *tet(A)*	*cib*, *traT*, *mer*	Positive
SB17032	Slaughterhouse	Thighs	2017	DK	350	+	pSB17032	85947bp	*bla* _CMY-2_	*-*	*traT*	Positive
SB17042	Slaughterhouse	Thighs	2017	DK	57	+	pSB17042	86724bp	*bla* _CMY-2_	*-*	*traT*	Positive
HBI01	Clinical ^c^	Blood	2017	DK	ST429	+	pHBI01	119331bp	*bla* _CMY-2_	*sul1*, *aac (3)-Via*, *aadA1*, *tet(A)*	*cib*, *traT*	Positive
HBI02	Clinical	Blood	2020	DK	ST69	+	pHBI02	85938bp	*bla* _CMY-2_	-	*traT*	Positive
HBI03	Clinical	Blood	2021	DK	ST131	+	pHBI03	85954bp	*bla* _CMY-2_	-	*traT*	Positive
HBI04	Clinical	Blood	2019	DK	ST95	+	pHBI04	98160bp	*bla* _CMY-2_	-	*traT*	Positive
HBI05	Clinical	Blood	2017	DK	ST12	+	pHBI05	96311bp	*bla* _CMY-2_	-	*traT*	Positive
2E151-R	FI farm ^d^	Egg	2017	FI ^b^	ST429							Positive
PS310-R	FI farm	Parent bird	2017	FI	ST429	+	pPS310-R	122067bp	*bla* _CMY-2_	*sul1*, *aac (3)-Via*, *aadA1*, *tet(A)*	*cib*, *traT*, *mer*	Positive
PS184-R	FI farm	Parent bird	2017	FI	ST429	+	pPS184-R	122085bp	*bla* _CMY-2_	*sul1*, *aac (3)-Via*, *aadA1*, *tet(A)*	*cib*, *traT*, *mer*	Positive
PS148-R	FI farm	Parent bird	2017	FI	ST429							Positive
PS378-R	FI farm	Parent bird	2017	FI	ST429							Positive
Y13-R	FI farm	Environment	2017	FI	ST429							Positive
PS57-R	FI farm	Parent bird	2017	FI	ST429							Positive
RL97	Food ^e^	Chicken Meat	2012	DE ^c^	ST429	+	pRL97	120705bp	*bla* _CMY-2_	*sul1*, *aac (3)-Via*, *aadA1*, *tet(A)*	*cib*, *traT*, *mer*	Positive
RL316	Food	Chicken Meat	2012	DE	ST429	+	pRL316	117209bp	*bla* _CMY-2_	*sul1*, *aac (3)-Via*, *aadA1*, *tet(A)*	*cib*, *traT*, *mer*	Positive
RL320	Food	Chicken Meat	2012	DE	ST429	+	pRL320	117209bp	*bla* _CMY-2_	*sul1*, *aac (3)-Via*, *aadA1*, *tet(A)*	*cib*, *traT*, *mer*	Positive

* Countries: DK: Denmark; FI: Finland; DE: Germany. ^a,b^—ESC-resistant *E. coli* isolates from Danish poultry farms and slaughterhouses, collected in a previously published study [20]. ^c^—Clinical ESC-resistant *E. coli* isolates obtained from the Statens Serum Institut in Denmark; ^d^—ESC-resistant *E. coli* isolates from Finnish poultry farms collected in a previously published study [22]. ^e^—ESC-resistant *E. coli* isolates from the German chicken meat, collected in a previously published study [14].

## Data Availability

All short-read sequences for the Danish isolates in this study had previously [20] been deposited with links to BioProject PRJNA1036573 in the NCBI BioProject. Additional raw data sequences used in this study were downloaded from the European Nucleotide Archive (ENA) under BioProjects PRJEB23663 (German) and PRJEB48023 (Norway). Accession numbers of each of the ST429 isolates were downloaded in the EnteroBases and provided in Appendix A. Complete hybrid assemblies of plasmid sequences from the present study were submitted to GenBank and can be assessed in BioProject PRJNA1049330 (https://www.ncbi.nlm.nih.gov/bioproject/, accessed on 5 March 2024).

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
