# Peer review of "Comparison of IncK-blaCMY-2 Plasmids in Extended-Spectrum Cephalosporin-Resistant Escherichia coli Isolated from Poultry and Humans in Denmark, Finland, and Germany"

_antibiotics, 2024, doi:10.3390/antibiotics13040349_

Round 1

Reviewer 1 Report

Comments and Suggestions for Authors

the research is very interesting and complete. just few comments:

in the intro, the EMA categorisation colud be mentioned.

the discussion is quite long , better to synthetize

MM: the selected isolates should be better described ( epidemiology). e.g. a table

Reviewer 2 Report

Comments and Suggestions for Authors

1.         I want to express my appreciation for the opportunity to review the manuscript titled "Comparison of IncK-blaCMY-2 plasmids in extended-spectrum cephalosporin-resistant Escherichia coli isolated from poultry and humans in Denmark, Finland, and Germany" for potential publication in Antibiotic. However, there are major concerns that need to be addressed to fully consider it for publication. 

2.         The introduction of the research presents and justifies its purpose in a general manner. It should provide a concise overview of the relevant literature that supports the study's hypotheses and aims. To enhance the clarity and coherence of the introduction, it is recommended that the author revise the final paragraph to offer a clearer outline of the research's purpose

3.         In the Materials and Methods section, it lacks sufficient content regarding the identification of Escherichia coli. Therefore, I kindly suggest that you provide more detailed information on how E. coli isolates were identified in your study.

4.         I have some concerns about the method of sample collection used in this study, which may have resulted in limited diversity in the obtained data. For example, the samples consisted of 4 parent bird samples from Finland in 2017, 3 chicken meat samples from Germany in 2012, and 12 meat and 4 intestine samples from a slaughterhouse in Denmark in 2017. This restricted spatio-temporal scope has the potential to impact the robustness of the epidemiological analysis conducted. It is important to acknowledge that the limited number and specific sources of the samples may introduce biases and restrict the generalizability of the findings. It would be beneficial to consider these limitations when interpreting the results and drawing conclusions from the study.

5.         I recommend presenting the experimental results in 2.2, which focus on the high prevalence of antimicrobial resistance determinants, or Figure 1, in the form of numerical patterns or cluster analysis.

6.         I kindly request clarification regarding the terms "Meat" and "Chicken Meat" in Table 2. It will enable a better understanding of their relationship or unique attributes within the context of the table.

7.         Furthermore, I would like to seek clarification regarding the term "Parent bird" in Table 2. Specifically, I would like to know which sample was collected in your study.

8.         I would recommend that the author thoroughly addresses the study's limitations within the discussion section. By acknowledging and discussing the limitations, the author can provide a more comprehensive and transparent interpretation of the findings, enhancing the overall quality of the manuscript.

9.         There is an inconsistency in the usage of "antimicrobial" and "antibiotic" throughout the text, especially in terms of "Antimicrobial resistance" and "antibiotic resistance". It is important to clarify which term is most suitable for this manuscript, ensuring a consistent and accurate use of terminology. 

10.   The conclusion section of the manuscript seems to contain general information. It is crucial to explicitly state the main findings derived from the substantial content obtained and provide concrete conclusions that offer valuable insights into clinical applications. Furthermore, it is advisable to include a dedicated section within the conclusion that outlines potential avenues for future research related to the current study. This will help to enhance the overall impact and relevance of the study's findings.

Reviewer 3 Report

Comments and Suggestions for Authors The main topic presented in the manuscript prepared by Che et al. is interesting.
In the study, the authors investigated the persistence, dynamics, and diversity of 46 ESC-resistant E. coli
and their IncK-blaCMY-2 plasmids, isolated from poultry and human samples, using both short-read and long-read sequencing.
The results revealed that different E. coli STs carried highly similar IncK-blaCMY-2 plasmids, with the most common ST429
being isolated from poultry and a human blood infection.
The idea and logical structure of the article are very clear, and is suitable
for publication in this magazine.

Author Response

Thank you for reviewing this paper and providing constructive suggestions. We also appreciate the supportive comments.

Round 2

Reviewer 2 Report

Comments and Suggestions for Authors

Thank you for your timely and comprehensive response to my comments on your manuscript. I want to express my gratitude for the dedication you have shown in addressing the concerns raised during the review process.

Upon careful consideration of your response and a thorough re-evaluation of the manuscript, I am pleased to say that your revisions have significantly enhanced the study. I firmly believe that it will make a meaningful and valuable contribution to the field.